# The Mediating Effect of Psychological Resilience between Individual Social Capital and Mental Health in the Post-Pandemic Era: A Cross-Sectional Survey over 300 Family Caregivers of Kindergarten Children in Mainland China

**Juxiong Feng [1], Pengpeng Cai [2], Xin Guan [3], Xuhong Li [4], Langjie He [1], Kwok-kin Fung [1] and Zheyuan Mai [5],***

1   Department of Social Work, Hong Kong Baptist University, Hong Kong, China;
    19481268@life.hkbu.edu.hk (J.F.); 20483198@life.hkbu.edu.hk (L.H.); kkfung@hkbu.edu.hk (K.-k.F.)
2   Trinity Centre for Global Health, School of Psychology, Trinity College Dublin, D02 PN40 Dublin, Ireland;
    caip@tcd.ie
3   Guangzhou Xinhua University, Dongguan 523133, China; guanxin@xhsysu.edu.cn
4   Department of Social and Behavioural Sciences, City University of Hong Kong, Hong Kong, China;
    xuhongli-c@my.cityu.edu.hk
5   Pentecostal Gin Mao Sheng Primary School, Hong Kong, China
*   Correspondence: zhe747659874@gmail.com

**Abstract:** In the context of the impact of the post-COVID-19 pandemic on families, this study explores the impact of individual social capital and psychological resilience on the mental health of family caregivers of kindergarten children in mainland China. This study included a sample of 331 family caregivers from Zhaoqing City, Guangdong Province, and the researchers applied the Personal Social Capital Scale (PSCS-16), Connor–Davidson Resilience Scale (CD-RISC-10), and Depression Anxiety Stress Scale (DASS) to assess social capital, psychological resilience, and mental health. Findings indicate a positive relationship between bridging social capital and mental health, while psychological resilience is negatively associated with depression, anxiety, and stress. Psychological resilience is identified as a mediator between social capital and mental health outcomes in this study. These insights highlight the importance of enhancing social capital and psychological resilience to improve family caregivers' mental health and the need for targeted interventions.

**Keywords:** social capital; caregivers; psychological resilience; depression; anxiety; stress

## 1. Introduction

The COVID-19 pandemic has significantly impacted family dynamics, particularly accentuating the challenges to family caregivers. Many parents have reported worsening mental health outcomes due to the increased stress and responsibilities during the pandemic (Calvano et al. 2021). Moreover, research has indicated that the negative outcomes of the pandemic to mental health outcomes not only manifest during the pandemic, but also lingering in the post-pandemic era (Costa et al. 2022). In this context, family caregivers assume a crucial role in the emotional and physical care of small children, offering comprehensive care that includes emotional support, practical assistance, and the protection of the child's health and well-being (Brodaty and Donkin 2009; Chadda 2014). In discussing the mental health of the family caregiver, social capital was explored as important because it contributes to the mental well-being (Lewis et al. 2013; Santini et al. 2015). While providing the caring work, family caregivers rely on social capital, including the resources, support, and connections they amass within their social networks (Furukawa and Greiner 2020; Mandelbaum et al. 2020). Social capital referring to "the sum of durable, trustworthy, reciprocal and resource-rich network connection" (Chen et al. 2009, p. 306) that facilitates

the caregivers with accessing to emotional support, practical assistance, and informational resources that help them navigate the challenges they face (Furukawa and Greiner 2020). It involves having a network of supportive peers, family members, and community members who provide guidance, understanding, and tangible assistance when needed.

Psychological resilience is essential for family caregivers. It refers to their ability to adapt, recover, and maintain their health in the face of adversity or duress (Palacio G et al. 2020). Family caregivers with greater levels of psychological resilience are better equipped to deal with the demands and challenges of caregiving (Giesbrecht et al. 2015; Roberts and Struckmeyer 2018). They are more likely to employ effective coping strategies, maintain a positive outlook, and protect their own mental health while fulfilling their caregiving duties (Roberts and Struckmeyer 2018).

In the Chinese context, the estimated prevalence of mental disorders among adults in China was reported to be 17.5% (Que et al. 2019). Among people with mental illness, the cultural principles of social obligation, reciprocity, and loyalty in China contribute to the expectation of significant rates of family care (Leng et al. 2019). Thus, it is also important to pay attention to the role of family caregivers and their mental health. Research suggests that social capital and psychological resilience have been identified as positive factors influencing the mental health of family caregivers in the Chinese context (Dai and Gu 2021; Su et al. 2021). Furthermore, in exploring the connection between social capital, psychological resilience, and mental health, numerous empirical researches support the argument that psychological resilience plays a mediating role between social capital and mental health (Adelinejad et al. 2022; Gao et al. 2018).

Therefore, this study is significant in understanding the role of social capital and psychological resilience in the mental health of family caregivers of children in Mainland China. The purpose of this study is to investigate the influence of individual social capital on the mental health of family caregivers and to explore the potential mediating effect of psychological resilience.

### 1.1. Social Capital and Mental Health

Numerous research studies have been conducted to explore the correlation between social capital and mental health (Poortinga 2006; Dominguez and Arford 2010; Ehsan et al. 2019). Empirical studies indicated that persons who possess a higher degree of social capital generally experience more positive mental health outcomes (Bassett and Moore 2013; Moore and Kawachi 2017), while some scholars argued that social capital has a negative relationship with health (Villalonga-Olives et al. 2017). On one hand, emotional support provided through social capital is instrumental in buffering against the negative effects of caregiving stress (Gazzaz et al. 2022; Mandelbaum et al. 2020; Oh and Park 2020). Specifically, different types of social capital, including bonding and bridging social capital, play an essential role in family caregivers' mental health (Roth 2020; Salehi et al. 2019). Bonding social capital refers to close and dense networks with members who have had long periods of interaction with each other (Putnam 2000; Horiuchi et al. 2013), and it contributes to better health through the positive effects of individual social networks and social support (Poortinga 2006). A supportive bonding network of friends and relatives who offer guidance, understanding, and tangible help can provide family caregivers with a sense of belonging, validation, and reduced isolation (Newman et al. 2019; Trail et al. 2020).

Bridging social capital refers to vertical networks that link social groups that differ demographically, economically, or hierarchically, thus allowing them to build consensus and exchange information (Putnam 2000; Rogers and Jarema 2015); it also bridges the resources beyond the family and community helping with the caring tasks for the caregivers (Barrett et al. 2014; Eberl 2020). Furthermore, the resources and caring services offered by the bridging social network could help reduce the caring burden linked with the lower levels of depression, anxiety, and caregiver burden among family caregivers (Barrett et al. 2014; Furukawa and Greiner 2020; Teahan et al. 2018). Thus, bonding and bridging social

capital could bring emotional support to caregivers and positively contribute to their mental health.

On the other hand, practical assistance is an essential component of social capital that directly impacts the mental health of family caregivers (Rosenberg et al. 2015). Practical support from different social networks can involve assistance with daily caregiving tasks and respite care or help navigating healthcare and support services. The availability of such support from the bonding network can alleviate the caregiver's workload, bridge social networks, provide opportunities for self-care, and reduce feelings of overwhelm and burnout, ultimately positively affecting their mental well-being (Rosenberg et al. 2015; Salehi et al. 2019).

In addition, informational resources within social networks also contribute to the mental health of family caregivers (Drouin et al. 2020). Access to accurate and relevant information about the child's condition, available resources, and coping strategies through both bonding and bridging networks can empower caregivers, enhance their decision-making abilities, and provide a sense of control and confidence in managing their caregiving responsibilities (Chen et al. 2015; Lin et al. 2013). This knowledge gained through social capital can mitigate anxiety, uncertainty, and feelings of incompetence, thereby promoting better mental health outcomes.

Social capital and its relationship with mental health have been extensively studied in Mainland China, shedding light on the cultural, social, and familial factors influencing individuals' mental well-being (He et al. 2019; Wu et al. 2012, 2014). In this context, social capital is deeply rooted in collectivism, interpersonal relationships, and reciprocal obligations (Herrmann-Pillath 2010). Family caregivers heavily rely on social networks, including extended family members, neighbours, and community support systems, to fulfil their caregiving responsibilities, and the availability of social support and resources within these networks significantly impact their mental health (Wu et al. 2012; Xu et al. 2021).

Previous research conducted in Mainland China has demonstrated that higher levels of social capital are associated with better mental health outcomes among family caregivers (Li et al. 2020; Wen and Lin 2012; Wu et al. 2015). Stronger social support networks and greater access to practical assistance have been consistently linked to lower caregiver burden, depression, and anxiety (Lai and Thomson 2011). The social connections and available resources within the Chinese cultural context act as protective factors, promoting resilience and overall well-being among family caregivers (Chan 2011). However, the mental health of family caregivers in conventional child-rearing environments has been neglected in prior research, which has concentrated chiefly on parents of children with special needs or migrant families. This study examines the mental health of family carers of kindergarten children without extra difficulties, offering a new viewpoint to the literature. The results will illuminate how social capital affects mental health in an understudied community.

### 1.2. Psychological Resilience and Mental Health

Resilience is a broad concept that refers to the capacity of an individual to adapt, recover, and flourish in the face of adversity, duress, or significant life challenges (Aburn et al. 2016; Windle 2011). It incorporates psychological, physiological, and social dimensions. Resilience is maintaining well-being, positive functioning, and a sense of equilibrium in adversity (Richardson and Chew-Graham 2016). Within the larger concept of resilience, psychological resilience concentrates explicitly on the psychological aspects of an individual's capacity to cope, recover, and thrive in the face of adversity (Graber et al. 2015; He et al. 2013; Ong et al. 2006). It accentuates the cognitive, emotional, and behavioral processes contributing to an individual's capacity to overcome obstacles, maintain mental health, and foster a positive development (He et al. 2013). Psychological resilience includes emotional regulation, self-efficacy, optimism, adaptive coping strategies, and utilizing personal strengths and resources (Li and Xie 2022; Süss and Ehlert 2020).

The mental health of individuals who care for children is significantly impacted by psychological resilience. Psychological resilience has a positive effect on the mental well-

being of caregivers (Ding et al. 2023). Research focusing on families caring for children with cancer (Toledano-Toledano et al. 2021) has found that higher levels of resilience are linked to enhanced quality of life, psychological well-being, and reduced levels of sadness, anxiety, and caregiver stress. Similarly, in the context of schizophrenia caregiving, caregivers with higher levels of psychological resilience experience less psychological distress, and psychological resilience moderates the relationship between stigma and psychological distress (Chen et al. 2016). In addition, a psychometric evaluation of the Mexican Measurement Scale of Resilience (RESI-M) among family caregivers of children with cancer revealed its reliability and construct validity, highlighting its usefulness in assessing psychological resilience in this population (Toledano-Toledano et al. 2019). Collectively, these findings highlight the positive impact of psychological resilience on caregivers' mental health and well-being, highlighting its function as a protective factor in mitigating the effects of caregiving stressors.

Psychological resilience is crucial for kindergarten-aged children's family caregivers who endure multiple stressors and demands in their caregiving roles (Jamison et al. 2023; Granek et al. 2014; Lee et al. 2021). These caregivers frequently face unique challenges related to time management, financial pressures, emotional distress, and balancing their requirements with those of their children. There is a considerable gap about the experiences of family caregivers of kindergarten-aged children, despite the fact that previous research has investigated the impact of psychological resilience on the mental health of caregivers in a variety of contexts. Few studies have investigated the role of psychological resilience in the population of family members who care for kindergarten-aged children, particularly in regard to the distinct challenges they encounter. This study seeks to address this lacuna in the literature by investigating the psychological resilience factors among kindergarten-aged children's families and their implications for these caregivers' mental health and well-being.

### 1.3. The Mediation Role of Psychological Resilience between Social Capital and Mental Health

Several empirical studies have supported the mediation role of psychological resilience between social capital and mental health (Adelinejad et al. 2022; Gao et al. 2018; Khaksar et al. 2019). A study conducted by Gao et al. (2018) in China revealed that persons with a greater degree of social capital, as measured among those living with HIV/AIDS, tend to have stronger psychological resilience, leading to improved mental health outcomes. Similarly, Adelinejad et al. (2022) revealed that social capital significantly influences university students' psychological resilience and mental health. Additionally, Khaksar et al. (2019) demonstrated that social capital plays a vital role in diminishing job burnout among employees in stressful settings, with psychological resilience acting as a facilitating role.

Further investigation into the correlation between social capital, stigma and psychological resilience revealed that Gao et al. (2018) found a negative link between stronger stigma, diminished levels of social capital, decreased psychological resilience, and increased risk of post-traumatic stress disorder (PTSD). This finding suggests that social capital has an impact on mental health outcomes through the reduction of stigma, in addition to its direct influence on psychological resilience. The impact of social capital on the mental well-being of university students has also garnered considerable attention (Gao et al. 2018). Adelinejad et al. (2022) conducted a study on university students and examined the impact of social capital on students' mental well-being among the COVID-19 pandemic. They specifically explored how resilience acts as a mediator in this relationship. The findings of the study revealed that social capital had a substantial influence on both psychological resilience and mental health outcomes. The results showed that psychological resilience partially mediated the relationship between social capital and mental health (Adelinejad et al. 2022). Based on the theoretical framework proposed by Adelinejad et al. (2022), which explores the mediating role of resilience between social capital and mental health, a research framework was developed for this study (Figure 1). Overall, these studies provide empirical evidence supporting the mediation role of psychological resilience between social capital and mental health outcomes.

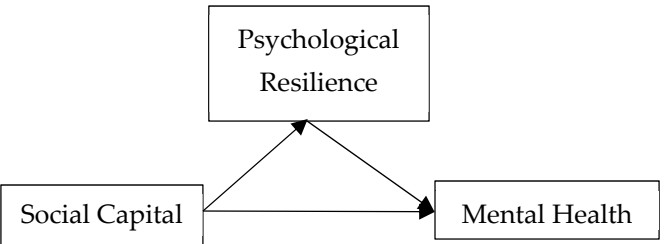

**Figure 1.** Theoretical Framework of the Research.

*1.4. The Current Study*

This study aims to comprehend the specific effect of social capital on the mental health of caregivers, and to examine the potential role of psychological resilience as a mediator between social capital and mental health among family caregivers of kindergarten-aged children in mainland China. Studies on family caregivers' mental health in mainland China have primarily focused on the caregivers of children with special needs (Chiu et al. 2013; Tong et al. 2022; Zhou et al. 2018) and old adults (Liu et al. 2019, 2022), but fewer studies have explored the mental health of caregivers of typically developing children in kindergarten. Thus, the kindergarten-aged children's family caregivers were chosen in this study. By investigating these associations, the study contributes to a better comprehension of the factors that influence the well-being of caregivers and provides insight into interventions and support programs.

## 2. Materials and Methods

*2.1. Participants and Sampling*

A public kindergarten located in the old urban area of Zhaoqing City was chosen as the research site to ensure caregiver diversity due to the diverse socioeconomic backgrounds of the family caregivers. A total of 331 family caregivers in the kindergarten were recruited based on the following criteria: (1) as a family caregiver of kindergarten children, (2) can be relied upon to answer the survey's question honestly, and (3) are above 18 years old. Regarding the participants, 38.3% identified as fathers, while 61.3% identified as mothers. The majority of the sample (96.1%) reported being married, and the average age of the participants was 37.68 years. A significant proportion (95.8%) of the participants were employed, and more than half reported a monthly income exceeding 5001 RMB, indicating a relatively high income level in Zhaoqing City.

The data collection process was supported by the kindergarten principal and teachers in March 2023. Firstly, the research teams provided a briefing on the research and questionnaire for the volunteer teachers from the kindergarten. Secondly, the recruited participants were invited to fill out the paper questionnaire in a face-to-face setting, with the distribution of the questionnaires being assisted by volunteer teachers. All participants volunteered to take part in the study and had the right to withdraw at any stage without facing any penalties. All research data are kept confidential and used solely for research purposes. The ethical approval was granted by the Research Committee of the Guangzhou Xinhua University (Approval Code: GZXH-IRB-20230201).

*2.2. Measurements*

2.2.1. Personal Social Capital Scale

This study employed a Personal Social Capital Scale (PSCS-16) to measure the social capital including bonding and bridging social capital (Wang et al. 2014). Bonding social capital was measured with the individual's embedment in different networks (such as family members, friends' network) while bridging social capital was explored by measuring the extent of interaction of an individual with different social organizations (Wang et al. 2014). The original PSCS consisted of 42 items and was validated as a reliable measure to

assess social capital (Chen et al. 2009) and have been in different culture context (Archuleta and Miller 2011).

There are 16 items on the PSCS-16 and each item is scored on a five point Likert scale ranging from 1 (a few) to 5 (all) to assess how many members you know in this network type. The overall score of the social capital scale is calculated by summing the responses for each item, ranging from 16 to 80. Specifically, the score for the bonding dimension is obtained by summing the responses for items 1 to 8, while the score for the bridging dimension is obtained by summing the responses for items 9 to 16. Each dimension's score ranges from 8 to 40; the higher the score, the higher the level of social capital. The validity and reliability of this questionnaire have been validated by a Cronbach's alpha coefficient of 0.965 in the Chinese context (Jiang et al. 2022). The scale applied in this study reported the high reliability, with a Cronbach's alpha of 0.92, for the overall social capital scale, and 0.882 and 0.887 for the bonding and bridging subscales, respectively.

### 2.2.2. Depression Anxiety Stress Scales

The present study employed a 21-item Depression Anxiety Stress Scale (DASS) to explore the presence of these symptoms in the past week, including depression, anxiety, and stress (Moussa et al. 2016). The DASS with a short 21-item form is a self-report instrument to explore the three related health states of depression, anxiety and stress (Antony et al. 1998; Lovibond and Lovibond 1995; Moussa et al. 2016). There are 7 items of those three subscales and each item used 4-point Likert-type scales to measure the amount to which individuals have experienced every single thing in the past week. Participants assigned a numerical value ranging from 0 (showing no correspondence) to 3 (representing a high degree of correspondence or frequency) to each question. The total score for the DASS is obtained by summing the responses to all 21 items, with a range of 0–63. Similarly, the total score for each subscale is calculated by summing the responses to the corresponding 7 items, resulting in a total score range of 0–21 points. Higher scores were indicative of a significant likelihood of experiencing mental health issues such as depression, anxiety, and stress.

The Chinese version of DASS has been validated in the context of Hong Kong demonstrating good internal consistency for each subscale (depression $\alpha = 0.903$, anxiety $\alpha = 0.776$, stress $\alpha = 0.864$) (Hue and Lau 2015). Additionally, the Chinese version of the DASS-21 also exhibited good internal consistency (Cronbach's alpha) with a value of 0.89 for the entire scale when applied in the context of mainland China (Gong et al. 2010). A test of reliability of DASS and three subscales revealed an excellent level in this study (overall DASS scale Cronbach's alpha = 0.96, depression $\alpha = 0.914$, anxiety $\alpha = 0.885$, and stress $\alpha = 0.871$).

### 2.2.3. Connor–Davidson Resilience Scale

The Connor–Davidson Resilience measure is a self-administered evaluation that assesses an individual's capacity to cope with stress. A 25-item scale was created using the principles of hardiness, adaptation, and stress tolerance, and its validity was confirmed by testing with various population groups (Connor and Davidson 2003). Modifications developed the 10-item Connor–Davidson Resilience Scale (CD-RISC-10) from the original version of 25 items, and the internal consistency was evaluated by calculating Cronbach's alpha. The alpha value of 0.85 indicated good reliability (Campbell-Sills and Stein 2007).

This study applied the Chinese version of the CD-RISC-10 (Ye et al. 2017). The validity and reliability of the CD-RISC-10 were confirmed with Cronbach's alpha, and were 0.92 and 0.90 for the clinical and non-clinical samples, respectively (Cheng et al. 2020). The 10-item scale is a self-administered questionnaire with a single dimension. It refers to five response options (0 = never; 4 = almost always) by using 5-point Likert-type scales. The total score of the scale is calculated by summing the responses for each item, which can vary from 0 to 40. Higher scores on the scale indicate a greater ability for resilience. The Cronbach's alpha coefficient of the scale used in the current study was 0.93.

*2.3. Data Analysis*

In terms of data analysis, we utilized SPSS version 26.0 and SPSS PROCESS version 4.3 to perform a statistical analysis. The purpose was to investigate the factors influencing mental health outcomes and examine the potential mediating role of psychological resilience in the relationship between social capital and mental health.

Initially, we conducted descriptive statistics to summarize data characteristics, followed by a Pearson's correlation analysis to explore relationships between variables. Subsequently, we employed hierarchical regression to investigate the influence on mental health outcomes by the sociodemographic factors, social capital, and psychological resilience. Sociodemographic factors (such as gender, income and education) were selected for a hierarchical regression analysis because those factors have a significant impact on caregivers' mental health outcomes (Penning and Wu 2016; Zhou et al. 2014). Finally, Hayes' (2018) PROCESS SPSS macro (Model 4) was utilized to explore the mediating effect of psychological resilience on the relationship between social capital and mental health outcomes. For each analysis, the *p*-value had to be lower than 0.05 to meet the criteria for the established statistical significance.

**3. Results**

*3.1. Characteristics of Participants*

In total, 331 participants completed the questionnaires, 38.3% of whom were fathers, 61.3% of whom were mothers. 96.1% of the sample were married, and the mean age was 37.68 years (SD = 5.50). The vast majority (95.8%) were employed. More than half reported a high monthly income (>5001 RMB). Table 1 displays the sociodemographic characteristics of participants.

**Table 1.** Sociodemographic Characteristics of Family Caregivers.

| | N (Mean) | % (SD) |
|---|---|---|
| Age | 37.68 | 5.5 |
| Gender | | |
| Father | 128 | 38.7 |
| Mother | 203 | 61.3 |
| Marital status | | |
| Married | 318 | 96.1 |
| Divorce | 13 | 3.9 |
| Educational level | | |
| Junior Secondary | 15 | 4.5 |
| Senior Secondary | 95 | 28.7 |
| Tertiary Education (Non-degree/Sub-degree) | 118 | 35.6 |
| Tertiary Education (Bachelor's degree) | 95 | 28.7 |
| Tertiary Education (Postgraduate degree) | 8 | 2.4 |
| Personal income (RMB) | | |
| ≤3000 | 9 | 2.7 |
| 3001–5000 | 121 | 36.6 |
| 5001–10,000 | 170 | 51.4 |
| >10,000 | 31 | 9.4 |
| Employment status | | |
| Employed | 317 | 95.8 |
| Unemployed | 14 | 4.2 |

*3.2. Correlations among Main Variables*

As shown in Table 2, a significant association between social capital and resilience was found (*p* < 0.01). Bridging was positively associated with mental health outcomes (*p* < 0.01). At the same time, resilience was negatively associated with all mental health outcomes (*p* < 0.01).

**Table 2.** Bivariate Correlations among Main Variables.

|  | 1 | 2 | 3 | 4 | 5 | 6 |
|---|---|---|---|---|---|---|
| 1. Bonding | 1 |  |  |  |  |  |
| 2. Bridging | 0.659 ** | 1 |  |  |  |  |
| 3. Resilience | 0.325 ** | 0.208 ** | 1 |  |  |  |
| 4. Stress | 0.009 | 0.191 ** | −0.548 ** | 1 |  |  |
| 5. Anxiety | 0.021 | 0.181 ** | −0.489 ** | 0.876 ** | 1 |  |
| 6. Depression | 0.034 | 0.208 ** | −0.497 ** | 0.857 ** | 0.912 ** | 1 |

** $p < 0.01$.

### 3.3. Influencing Factors of Mental Health Outcomes

According to the hierarchical regression analysis shown in Table 3, age was positively relevant to stress (β = 0.129, $p$ = 0.003), anxiety (β = 0.187, $p < 0.001$) and depression (β = 0.171, $p < 0.001$). In terms of social capital, bridging was positively relevant to stress (β = 0.321, $p < 0.001$), anxiety (β = 0.291, $p < 0.001$) and depression (β = 0.328, $p < 0.001$). However, no significant relationship was found between bonding and mental health outcomes. Psychological resilience was negatively relevant to stress (β = −0.569, $p < 0.001$), anxiety (β = −0.501, $p < 0.001$) and depression (β = −0.514, $p < 0.001$). Additionally, it was observed that those with a higher income were more likely to suffer from mental health issues.

**Table 3.** Influencing Factors of Mental Health Outcomes.

|  | Stress | | | | Anxiety | | | | Depression | | | |
|---|---|---|---|---|---|---|---|---|---|---|---|---|
|  | Model 1 | | Model 2 | | Model 1 | | Model 2 | | Model 1 | | Model 2 | |
|  | β | $p$ | β | $p$ | β | $p$ | β | $p$ | β | $p$ | β | $p$ |
| Gender | 0.121 | 0.019 | 0.04 | 0.339 | 0.101 | 0.047 | 0.031 | 0.485 | 0.11 | 0.029 | 0.039 | 0.360 |
| Age | 0.259 | 0.000 *** | 0.129 | 0.003 ** | 0.301 | 0.000 *** | 0.186 | 0.000 *** | 0.291 | 0.000 *** | 0.171 | 0.000 *** |
| Marital status | −0.08 | 0.090 | −0.04 | 0.354 | −0.08 | 0.123 | −0.04 | 0.409 | −0.05 | 0.259 | −0.010 | 0.802 |
| Educational level | −0.2 | 0.000 *** | −0.05 | 0.247 | −0.17 | 0.000 *** | −0.05 | 0.321 | −0.18 | 0.000 *** | −0.046 | 0.291 |
| Personal income | 0.107 | 0.037 | 0.117 | 0.005 ** | 0.118 | 0.020 * | 0.127 | 0.004 * | 0.187 | 0.000 *** | 0.197 | 0.000 *** |
| Employment status | 0.009 | 0.848 | 0.042 | 0.289 | −0.01 | 0.919 | 0.024 | 0.557 | −0.03 | 0.523 | 0.000 | 0.995 |
| Bonding |  |  | −0.02 | 0.688 |  |  | −0.02 | 0.716 |  |  | −0.043 | 0.459 |
| Bridging |  |  | 0.321 | 0.000 *** |  |  | 0.291 | 0.000 *** |  |  | 0.328 | 0.000 *** |
| Psychological Resilience |  |  | −0.57 | 0.000 *** |  |  | −0.5 | 0.000 *** |  |  | −0.514 | 0.000 *** |
| $R^2$ | 0.132 | | 0.439 | | 0.149 | | 0.389 | | 0.167 | | 0.429 | |
| F for $R^2$ change | 9.46 *** | | 67.326 *** | | 10.898 *** | | 48.353 *** | | 12.469 *** | | 56.512 *** | |

* $p < 0.05$. ** $p < 0.01$. *** $p < 0.001$.

### 3.4. Mediation Analysis

The results (Figures 2–7) demonstrated the mediation effect of psychological resilience. Obviously, the indirect effect of bonding on stress (indirect effect = −0.1328, 95% CI [−0.1820, −0.0888]), anxiety (indirect effect = −0.1189, 95% CI [−0.1653, −0.0780]) and depression (indirect effect = −0.1377, 95% CI [−0.1909, −0.0915]) were significant. The indirect effect of bridging on stress (indirect effect = −0.0727, 95% CI [−0.1059, −0.0383]), anxiety (indirect effect = −0.0648, 95% CI [−0.0939, −0.0356]) and depression (indirect effect = −0.0752, 95% CI [−0.1097, −0.0412]) were significant. In addition, the direct effects of both bonding and bridging social capital on mental health were significant ($p < 0.001$). Therefore, psychological resilience was found to partially mediate the relationship between social capital and mental health outcomes.

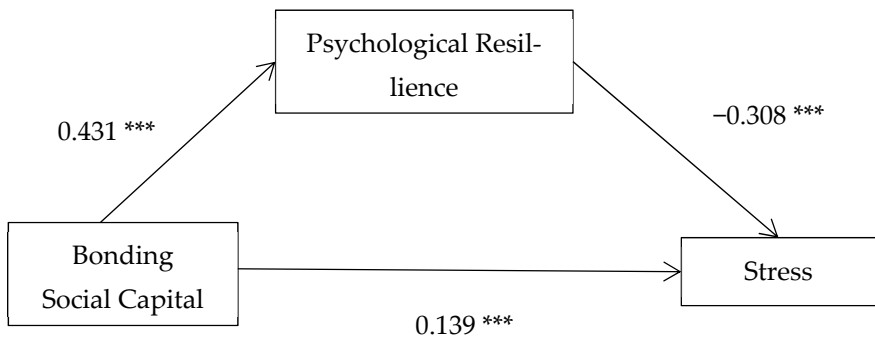

**Figure 2.** The Mediated Effect of Psychological Resilience between Bonding Social Capital and Stress. *** *p* < 0.001.

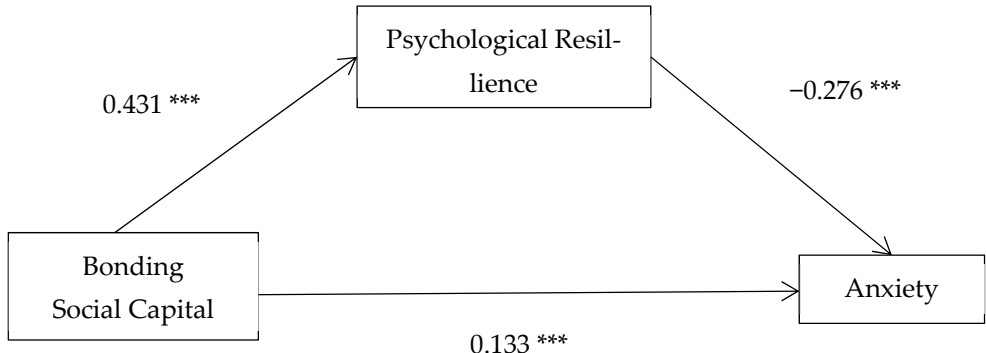

**Figure 3.** The Mediated Effect of Psychological Resilience between Bonding Social Capital and Anxiety. *** *p* < 0.001.

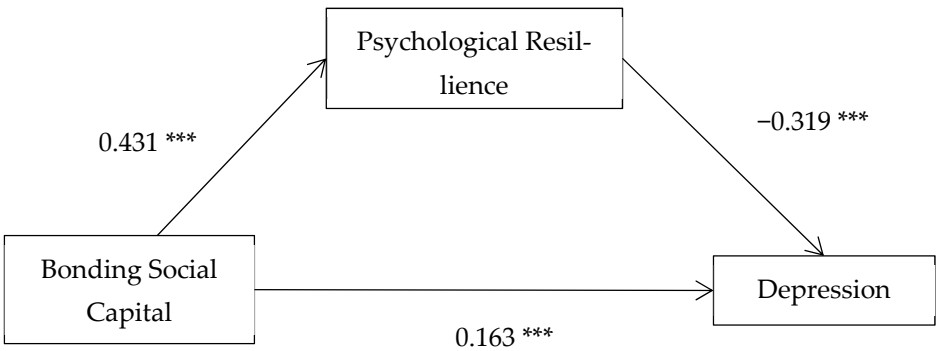

**Figure 4.** The Mediated Effect of Psychological Resilience between Bonding Social Capital and Depression. *** *p* < 0.001.

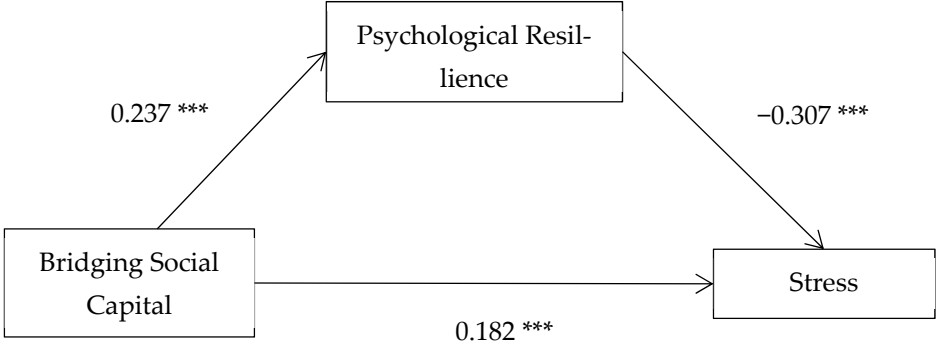

**Figure 5.** The Mediated Effect of Psychological Resilience between Bridging Social Capital and Stress. *** *p* < 0.001.

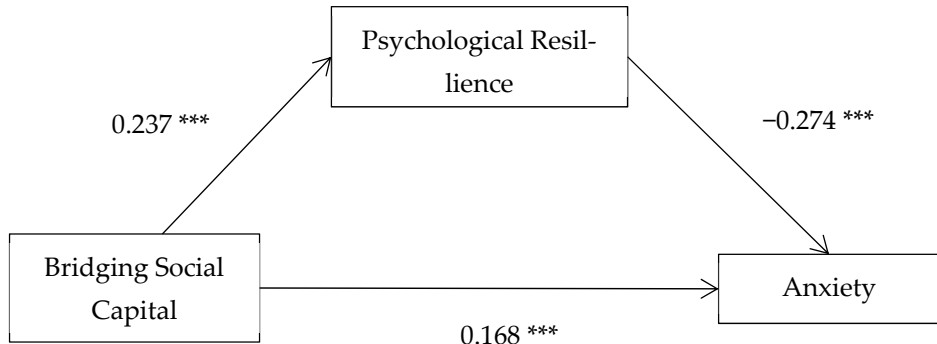

**Figure 6.** The Mediated Effect of Psychological Resilience between Bridging Social Capital and Anxiety. *** *p* < 0.001.

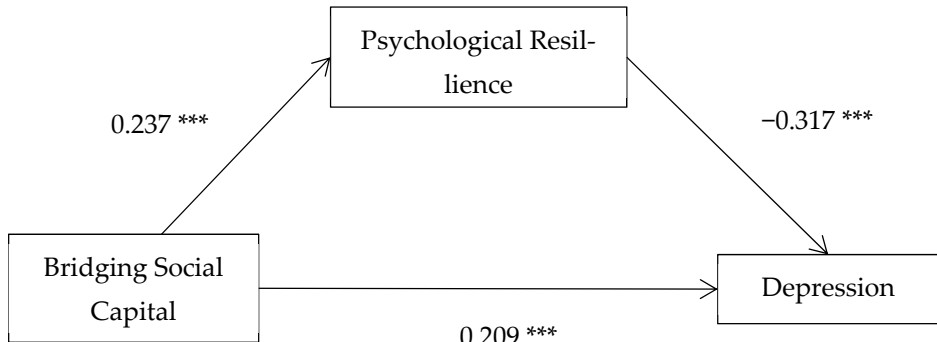

**Figure 7.** The Mediated Effect of Psychological Resilience between Bridging Social Capital and Depression. *** *p* < 0.001.

## 4. Discussion

The research showed that bridging social capital was positively related to mental health status, while psychological resilience was negatively related to anxiety, depression, and stress. It means that the family caregiver with a lower level of bridging social capital and a higher level of psychological resilience tends to have better mental health. This research also found that psychological resilience can mediate the relationship between social capital and mental health outcomes.

### 4.1. Social Capital and Mental Health

The debate on social capital and mental health is complex (Sartorius 2003). Some studies have pointed out a positive relationship between social capital and mental health, implying that social capital contributes to better mental status (Sun and Lu 2020; Xu et al. 2021). However, other scholars have also indicated the negative relationship between social capital and mental health. In the family caregiver group, Zhong et al. (2020) showed that social capital with a social support is negatively related to family caregivers' depressive symptoms and caregiver burden. Our research results also revealed that the higher the level of the bridging social capital, the higher the mental health score; that is, the worse the stress, anxiety, and depression of the caregivers. This also echoes previous research (Zhong et al. 2020). Recent studies indicated that specific forms of social capital can have detrimental effects on mental health, including bridging social capital (Ferlander et al. 2016; Chen et al. 2018). A study conducted to explore the relationship between social capital and depression among women revealed that women with higher levels of bridging capital, characterized by an increased interaction with individuals from different age groups outside the family, were more likely to report the high level of depression (Ferlander et al. 2016). Ferlander et al. (2016) further explained that this association could be attributed to workplace dynamics, which commonly involve generational conflicts over economic and cultural resources. These conflicts arise due to divergent work values across generations, potentially leading

to interpersonal conflicts and subsequent negative mental health outcomes. Similarly, in our study, a significant proportion of family caregivers were employed, indicating a higher level of bridging capital. However, the workplace environment can also lead to increased stress and depression among caregivers, especially when they have to balance caregiving responsibilities for their children and dual workloads. These factors can significantly impact their mental health outcomes, leading to negative consequences.

Bonding social capital plays a crucial role in promoting positive mental health outcomes, which facilitate the exchange of emotional support and instrumental support during challenging and stressful events (Salehi et al. 2014). Bonding social capital is commonly observed within close-knit social ties, such as family members that foster a sense of belonging, trust, and mutual support, contributing to improved mental well-being (Salehi et al. 2019). Thus, bonding capital could contribute to the caregiver's mental health. However, in our study, there is no statistically significant correlation between bonding capital and mental health. A study pointed out that non-caregivers had higher social capital scores compared to caregivers, with significant differences between those two groups (Papastavrou et al. 2015). When comparing caregivers and non-caregivers, it is noteworthy that the positive impact of bonding social capital on mental health may not always be substantial. When caregivers' high social capital is reflected in bonding network support, they tend to convene and engage in discussions regarding stress and anxiety. However, without the aid of external resources and support, their situation remains unresolved and impedes potential alleviation and improvement. Qualitative research is needed to explore their possibilities further.

### 4.2. Psychological Resilience and Mental Health

Our research reveals a direct relationship between psychological resilience and mental health among family caregivers. Specifically, increased psychological resilience was associated with lower DASS scores. The findings are consistent with those of Connor and Davidson (2003) who emphasized resilience as an effective buffer against various psychiatric symptoms. Also, psychological resilience is a protective factor that enables individuals to maintain or regain mental health in adversity (Rutter 1985). In addition, Windle (2011) argued that psychological resilience is a mechanism that moderates the adverse effects of stress and adversity on mental health, thus becoming a critical factor in preventing caregivers from developing mood disorders.

Research in recent years has also emphasized psychological resilience as a key buffer against mental health disorders, especially in situations of chronic stress or adversity, such as caregiving. For example, Vinkers et al. (2014) emphasized that psychological resilience may modulate neural and hormonal responses to stress, which in turn protects individuals from mental illness. Similarly, in the caregiving scenario, Ali et al. (2016) found that caregivers with greater psychological resilience exhibited fewer depressive symptoms, emphasizing the protective role of resilience against psychological distress.

Additionally, this study found an inverse relationship between psychological resilience and age. This may mean that, although psychological resilience is generally thought of as a more stable trait, younger caregivers may exhibit greater psychological resilience in some respects. This may be because they have been exposed to more modern understandings and intervention strategies of mental health, or their adaptive coping mechanisms are more in line with contemporary mental health concepts. According to Masten (2001), psychological resilience is the result of the interaction between the individual and the environment, and this interaction can change throughout the individual's life. Therefore, although psychological resilience is stable to a certain extent, it is also affected by one's life experiences and environment.

### 4.3. Social Capital and Psychological Resilience

The symbiotic relationship between social capital and psychological resilience plays a crucial role in an individual's well-being, especially in high-stress roles such as family

guardians or primary caregivers of young children. The interaction of these two factors can determine how effectively a person copes with adversity and meets challenges.

Recent research has delved into the mechanisms by which social capital works to increase psychological resilience. Social capital provides access to resources and information, fosters a sense of belonging, and serves as a platform for emotional support. For example, in a study exploring the role of social capital in community resilience after a natural disaster, Aldrich and Meyer (2015) found that communities with stronger social networks, trust, and shared norms (indicative of higher social capital) showed better resilience and recovery.

Applying this understanding to family guardianship research, family members with significant social capital have access to a variety of resources that not only directly assist them in their guardianship duties but also help build psychological resilience. Li et al. (2014) found in their study that caregivers with strong social networks reported better mental health outcomes, suggesting that social capital plays a protective role against potential psychological distress.

The reciprocal nature of this relationship is also of interest. When caregivers rely on their social networks, they receive support and contribute to strengthening those networks, thereby reinforcing the value and effectiveness of their social capital. Aldrich (2012) emphasizes this point, illustrating how caregivers can build deeper connections within their communities through shared experiences, which enhances collective social capital and boosts collective mental health status.

### 4.4. Mediating Role of Psychological Resilience

The complex interplay between social capital, psychological resilience, and specific dimensions of mental health in family caregivers is illuminated by the mediation model presented in the study. The research provides evidence that psychological resilience acts as a mediator between two types of social capital (bonding and bridging) and three specific variables of mental health: depression, anxiety, and stress.

Based on the theoretical framework and research results, there are three underlying mechanisms of the relationships between variables, including social capital and psychological resilience, social capital and mental health, and psychological resilience and mental health. Firstly, for social capital and psychological resilience, it is worth noting the pivotal role of bonding and bridging social capital in enhancing resilience. Psychological resilience has been increasingly recognized as a byproduct of resources and support obtained from social capital (Panter-Brick and Leckman 2013). Despite their emotionally and physically taxing responsibilities, family caregivers benefit immensely from a reservoir of social capital that provides tangible resources and emotional and psychosocial support.

Secondly, while previous research has shown that social capital contributes to better mental health outcomes by providing increased social support and fostering social connections (Adelinejad et al. 2022), our study reveals that social capital can have negative effects on mental health, especially bridging social capital. This result of this research is consistent with the study of Zhong et al. (2020) and Ferlander et al. (2016). Ferlander et al. (2016) indicated that people with relative higher level of bridging capital need to face the workplace conflict so that leading to poor mental health. In our study, the employed family caregivers, indicating a higher level of bridging capital, faced increased stress and depression due to balancing caregiving responsibilities and workloads. The workplace environment played a role in contributing to these negative mental health outcomes, particularly when caregivers had to manage both their children's needs and work demands.

Thirdly, psychological resilience contributed to better mental health in this study, consistent with previous empirical studies (Vinkers et al. 2014; Ali et al. 2016). Diving deeper into the mediation model's findings, psychological resilience appears to act as a buffer, converting the benefits of social capital into enhanced mental well-being by notably decreasing anxiety and depression levels. This aligns with earlier studies that highlight psychological resilience's role in offering protection against the development of anxiety and depressive disorders (Smith et al. 2008). It is imperative, therefore, to view

psychological resilience as a pivotal linchpin, tying together the resources from social capital and improved mental health outcomes for anxiety and depression.

Wei et al. (2022) suggest that although social capital and resilience jointly impact anxiety and depression among family caregivers, stress in this population might be influenced by other variables or may inherently resist mediation by resilience. This study highlights the mediating role of psychological resilience between social capital and stress, emphasizing the significance of comprehending psychological resilience. This nuanced comprehension underscores the multifaceted aspect of mental health and suggests the different potential needs for targeted interventions.

## 5. Implications

The implications of this study are multifaceted and may include implications for policy development, in-person caregiving practice, and future research. A central finding of this study is the undeniable importance of social capital in the home environment. Notably, family caregivers who are in rich social networks exhibit higher levels of resilience (Putnam 2000). Given the strong correlation between social capital and reduced indicators of anxiety and depression, there is a strong case for focusing interventions on fostering these networks for caregivers (Bourdieu 2008). This argument suggests that agencies and caregiver support groups could invest in programs that foster such connections within and beyond the caregiver community.

While numerous studies highlight the positive impact of social capital on mental health (Salehi et al. 2019; Adelinejad et al. 2022), this study serves as a reminder to consider the potential negative aspects of social capital, particularly within family caregiving groups. Balancing work and caregiving responsibilities can transform high levels of bridging capital into a burden, thereby influencing mental health. It is crucial for social services to be aware of the potential dark sides of bridging capital and address them accordingly. In addition, researches have shown that psychological resilience is an important mediator between social capital and certain aspects of mental health. This relationship coincides with the findings of Zimmerman et al. (2013), implying that increasing psychological resilience can help alleviate the mental health challenges caregivers often face. The benefits of resilience-building programs such as positive thinking exercises or training courses in social capital resources cannot be underestimated (Southwick et al. 2016).

Interestingly, the inverse relationship between psychological resilience and age suggests that older caregivers may be uniquely vulnerable. Given that psychological resilience may diminish with age, this emphasizes the need for age-sensitive interventions (Windle 2011). Different strategies or resources for different age groups would be more effective.

## 6. Limitation and Future Research

In the context of this study on the impact of social capital and psychological resilience on mental health, one limitation is that the cross-sectional design lacks the ability to make causal inferences. The cross-sectional study should also pay attention to recall bias and potential inaccuracies in self-report data. Thus, longitudinal research is expected to explore the dynamic interplay between social capital, psychological resilience, and mental health, which helps examine how changes in social capital and psychological resilience precede changes in mental health. Another limitation is that this research primarily focuses on individual social capital, which may obscure the subtle interplay of collective or community forms of social capital. Community social capital, including community participant, trust and reciprocity, should be considered (Lu et al. 2021; Roxas and Azmat 2014), as such a form of social capital can also contribute to better mental well-being (Wind and Komproe 2012). This broadened perspective has the potential to provide more comprehensive insights into the dynamics of caregivers' mental health. Looking to the future, there is an urgent need for more in-depth research into the multi-dimensional domain of social capital. Research exploring the differences between individual and collective social capital and their respective impacts on mental health could illuminate. In addition, the dynamics of

community resilience, particularly its impact on the mental health of caregivers of young children in families, could be examined in depth. In terms of practice, future research could be directed at designing and testing interventions that harness the power of social capital tools and resilience-building strategies to improve caregivers' mental health.

**Author Contributions:** Conceptualization, J.F. and P.C.; methodology, X.G.; software, X.L.; validation, J.F., P.C. and Z.M.; formal analysis, X.L.; investigation, X.G.; resources, Z.M.; data curation, L.H.; writing—original draft preparation, P.C. and J.F.; writing—review and editing, J.F. and Z.M.; visualization, L.H.; supervision, K.-k.F.; project administration, Z.M.; funding acquisition, X.G. All authors have read and agreed to the published version of the manuscript.

**Funding:** This research was funded by Guangzhou Xinhua College Higher Education Teaching Reform Project (General Category), grant number 2023J054, and the College-level Research Project (2019KYQN06) of Guangzhou Xinhua University.

**Institutional Review Board Statement:** The study was conducted in accordance with the Declaration of Helsinki, and the studies involving human participants were reviewed and approved by the Research Committee of the Guangzhou Xinhua University. (protocol code GZXH-IRB-20230201, 1 February 2023).

**Informed Consent Statement:** Informed consent was obtained from all participants involved in the study.

**Data Availability Statement:** Data sharing is unavailable due to reasons of privacy. Concerning inquiries pertaining to the data presented in this study, please contact the corresponding author.

**Conflicts of Interest:** The authors declare no conflict of interest.

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
