# Peer review of "The Mediating Effect of Psychological Resilience between Individual Social Capital and Mental Health in the Post-Pandemic Era: A Cross-Sectional Survey over 300 Family Caregivers of Kindergarten Children in Mainland China"

_socsci, doi:10.3390/socsci13020122_

Round 1

Reviewer 1 Report

Comments and Suggestions for Authors

The title should mention the context and the sample size of the study.

In the title, the authors present the mediating role of resilience but in the abstract, they wrote: “Psychological resilience is identified as a moderator between social capital and mental health outcomes in this study. Please, clarify this point.

Please, define the concept of social capital early, in the introduction section.

The authors describe the concept of well-being in the place of social capital in the introduction section. They should clarify the relationship between social capital and well-being, emphasizing both differences and communalities.

The authors should include a theoretical framework able to describe why and how social capital is linked with resilience.

The authors should explain why mental health is specifically study in terms of depression. Please, clarify the connection between social capital, mental health and resilience in the introduction.

The authors should detail information on the characteristics of the sample in the “participant and sampling section” instead of the results section.

The data analysis section was lacking. Please, provide a thorough explanation of the statistical method performed.

The authors should report the Cronbach alfa values for the bonding and bridging subscales of the social capital scale.

Please, present the Cronbach alfa values for stress, anxiety and depression subscales for the depression scale.

The authors should elaborate on why bonding Is negatively related with mental health and why bridging is positively related.

The authors should clarify the calculation method for social capital and depression score.

The authors should present the results of the study considering the sub-dimension of each scale. 

I expect observing both bridging and bonding sub-dimensions of social capital as antecedent of resilience and depression in the mediating model.

The authors are encouraged to integrate pertinent sociodemographic characteristics of Family Caregivers into the hierarchical regression, accompanied by a table. An explanation should accompany the rationale for selecting these variables.

Please, discuss the results, the implications of the findings, and elucidate the underlying mechanisms of the relationships between variables basing on the theoretical framework previously mentioned (as I have requested above).

The limitation and future research section should be extended and discussed in detail.

By addressing these points, the manuscript will enhance the overall quality.

Reviewer 2 Report

Comments and Suggestions for Authors

This work addresses the relationship between social capital and mental health, analyzing the mediating role of psychological resilience. The research is supported both empirically and theoretically. The authors have carried out a thorough analysis of the previous scientific literature, consulting recent and previous research. The methodology is correctly focused, as well as the statistical analyses. For its part, the discussion includes the contributions of the theoretical framework and future lines of intervention are proposed, as well as the limitations of cross-sectional research in studies of these characteristics. The following are suggested as suggestions for improvement to the manuscript:

Abstract:

Line 12 change the word moderator to mediator. Resilience acts as a mediator in the relationship between social capital and psychological resilience, which means that it explains that relationship.

Introduction:

I have missed both in the introduction (for example in section 1.1.) and in the discussion that there is no reference to the quality of the guidelines in relation to social capital (in section 4.1.). Perhaps it could explain some non-significant relationships obtained in the data analysis.

Method

Line 215: Include the code provided by the university bioethics committee.

Line 218: It is not necessary to put the initials of the instrument in parentheses in the title of subsection 2.2.1. Proceed in the same way on line 233, for subsection 2.2.2.

In figures 1, 2 and 3, specify the type of social capital in the social capital box.

Round 2

Reviewer 1 Report

Comments and Suggestions for Authors

The authors have addressed all of my questions, contributing to an enhanced quality of the manuscript. I recommend including in the results section whether the mediation role of Resilience is deemed full or partial. After making these slight modifications, I propose the publication of the manuscript.
